Journal of Data-centric Machine Learning Research (2024)          Submitted 3/24; Revised 5/24; Published 6/24

# You can't handle the (dirty) truth:
# Data-centric insights improve pseudo-labeling

**Nabeel Seedat**[*]                                                        NS741@CAM.AC.UK
*University of Cambridge, Cambridge, UK*

**Nicolas Huynh**[*]                                                       NVTH2@CAM.AC.UK
*University of Cambridge, Cambridge, UK*

**Fergus Imrie**                                                           IMRIE@UCLA.EDU
*University of California, Los Angeles, CA, USA*

**Mihaela van der Schaar**                                                 MV472@CAM.AC.UK
*University of Cambridge, Cambridge, UK*

**Reviewed on OpenReview:** *https: // openreview. net/ forum? id= 2tBwcT9z55*

**Editor:** Sergio Escalera

## Abstract

Pseudo-labeling is a popular semi-supervised learning technique to leverage unlabeled data when labeled samples are scarce. The generation and selection of pseudo-labels heavily rely on labeled data. Existing approaches implicitly assume that the labeled data is gold standard and "perfect". However, this can be violated in reality with issues such as mislabeling or ambiguity. We address this overlooked aspect and show the importance of investigating *labeled data* quality to improve *any* pseudo-labeling method. Specifically, we introduce a novel data characterization and selection framework called DIPS to extend pseudo-labeling. We select useful labeled and pseudo-labeled samples via analysis of learning dynamics. We demonstrate the applicability and impact of DIPS for various pseudo-labeling methods across an extensive range of real-world tabular and image datasets. Additionally, DIPS improves data efficiency and reduces the performance distinctions between different pseudo-labelers. Overall, we highlight the significant benefits of a data-centric rethinking of pseudo-labeling in real-world settings.

**Keywords:**   pseudo-labeling, semi-supervised learning, data characterization

## 1 Introduction

Machine learning heavily relies on the availability of large numbers of annotated training examples. However, in many real-world settings, such as healthcare and finance, collecting even limited numbers of annotations is often either expensive or practically impossible. Semi-supervised learning leverages unlabeled data to combat the scarcity of labeled data (Zhu, 2005; Chapelle et al., 2006; van Engelen and Hoos, 2019). Pseudo-labeling is a prominent semi-supervised approach applicable across data modalities that assigns pseudo-labels to unlabeled data using a model trained on the labeled dataset. The pseudo-labeled data is then

---

*. Equal Contribution

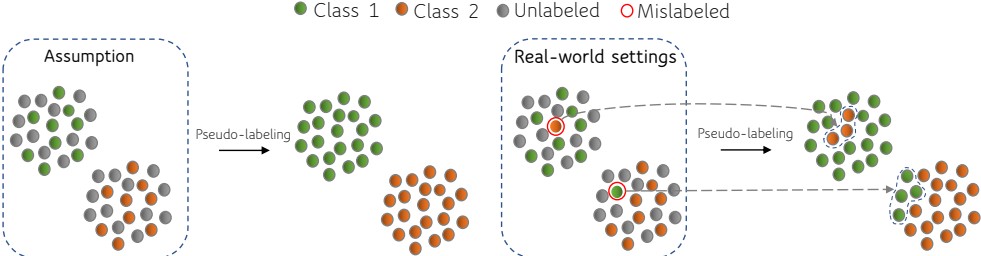

Figure 1: **(Left)** Current pseudo-labeling formulations implicitly assume that the labeled data is the gold standard. **(Right)** However, this assumption is violated in real-world settings. Mislabeled samples lead to error propagation when pseudo-labeling the unlabeled data.

combined with labeled data to produce an augmented training set. This increases the size of the training set and has been shown to improve the resulting model. In contrast, consistency regularization methods (Sohn et al., 2020) are less versatile and often not applicable to settings such as tabular data, where defining the necessary semantic-preserving augmentations proves challenging (Gidaris et al., 2018; Nguyen et al., 2022a). Given the broad applicability across data modalities and competitive performance, we focus on pseudo-labeling approaches.

**Labeled data is not always gold standard.** Current pseudo-labeling methods focus on unlabeled data selection. However, an equally important yet *overlooked* problem is around labeled data quality, given the reliance of pseudo-labelers on the labeled data. In particular, it is often implicitly assumed that the labeled data is "gold standard and perfect". This "gold standard" assumption is unlikely to hold in reality, where data can have issues such as mislabeling and ambiguity (Sambasivan et al., 2021; Renggli et al., 2021; Jain et al., 2020; Gupta et al., 2021a,b; Northcutt et al., 2021a,b). For example, Northcutt et al. (2021b) quantified the label error rate of widely-used benchmark datasets, reaching up to 10%, while Wei et al. (2022a) showed this can be as significant as 20-40%. This issue is critical for pseudo-labeling, as labeled data provides the supervision signal for pseudo-labels. Hence, issues in the labeled data will affect the pseudo-labels and the predictive model (see Fig. 1). Mechanisms to address this issue are essential to improve pseudo-labeling. It might appear possible to manually inspect the data to identify errors in the labeled set. However, this requires domain expertise and is human-intensive, especially in modalities such as tabular data where inspecting rows in a spreadsheet can be much more challenging than reviewing an image. In other cases, updating labels might be infeasible due to rerunning costly experiments in domains such as biology and physics, or indeed impossible due to lack of access to either the underlying sample or equipment.

**Extending the pseudo-labeling machinery.** To solve this fundamental challenge, we propose a novel framework to extend the pseudo-labeling machinery called **D**ata-centric **I**nsights for **P**seudo-labeling with **S**election (**DIPS**). DIPS focuses on the labeled and pseudo-labeled data to characterize and select the most useful samples. We instantiate DIPS based on learning dynamics — the behavior of individual samples during training. We analyze the dynamics by computing two metrics, confidence and aleatoric (data) uncertainty, which enables the characterization of samples as *Useful* or *Harmful*, guiding sample selection for model training. Sec. 5 empirically shows that this selection improves pseudo-labeling performance in multiple real-world settings.

Beyond performance, `DIPS` is also specifically designed to be a flexible solution that easily integrates with existing pseudo-labeling approaches, having the following desired properties:

**(P1) Plug & Play:** applicable on top of *any* pseudo-labeling method (to improve it).
**(P2) Model-agnostic data characterization:** agnostic to any class of supervised backbone models trained in an iterative scheme (e.g. neural networks, boosting methods).
**(P3) Computationally cheap:** minimal computational overhead to be practically usable.

**Contributions:** ① *Conceptually*, we propose a rethinking of pseudo-labeling, demonstrating the importance of characterizing and systematically selecting data from both the labeled and pseudo-labeled datasets, in contrast to the current focus *only* on the unlabeled data. ② *Technically*, we introduce `DIPS`, a novel framework to characterize and select the most useful samples for pseudo-labeling. This extends the pseudo-labeling machinery to address the unrealistic status quo of considering the labeled data as gold standard. ③ *Empirically*, we show the value of taking into account labeled data quality, with DIPS's selection mechanism improving various pseudo-labeling baselines, both in terms of *performance* and *data efficiency*, which we demonstrate across 18 real-world datasets, spanning both tabular data and images. This highlights the usefulness and applicability of `DIPS`.

## 2 Related work

**Semi-supervised learning and pseudo-labeling methods.** Semi-supervised learning leverages unlabeled data to combat the scarcity of labeled data (Zhu, 2005; Chapelle et al., 2006; van Engelen and Hoos, 2019; Iscen et al., 2019; Berthelot et al., 2019). As mentioned in Sec. 1, we focus on pseudo-labeling approaches, given their applicability across data modalities and competitive performance. Recent methods have extended pseudo-labeling by modifying the selection mechanism of unlabeled data (Lee et al., 2013; Rizve et al., 2021; Nguyen et al., 2022a; Tai et al., 2021), using curriculum learning (Cascante-Bonilla et al., 2020), or merging pseudo-labeling with consistency loss-focused regularization (Sohn et al., 2020). A commonality among these works is a focus on ensuring the correct selection of the unlabeled data, assuming a gold standard labeled data. In contrast, `DIPS` addresses the question: "What if the labeled data is not gold standard?", extending the aforementioned approaches to be more performant.

**Self-supervised learning.** In addition to semi-supervised learning, there exist other paradigms to leverage unlabeled data. For example, self-supervised learning is a popular technique to learn representations from large unlabeled datasets. It has been widely used in different modalities, e.g. computer vision (Oquab et al., 2023; Henaff, 2020; Chen et al., 2020), text (Devlin et al., 2019; Radford et al., 2019) and tabular data (Yoon et al., 2020; Lee et al., 2021). Similarly to consistency regularization, self-supervised learning is often specific to each modality, typically requires large quantities of unlabeled data, and only incorporates labeled data separately (e.g. during fine-tuning). This contrasts pseudo-labeling, which is a versatile and general semi-supervised approach applicable across modalities, and which uses labeled data in combination with unlabeled data.

**Data-centric AI.** Data-centric AI has emerged focused on developing systematic methods to improve the quality of data (Liang et al., 2022; Seedat et al., 2023b). One aspect is to

score data samples based on their utility for a task, or whether samples are easy or hard to learn (Seedat et al., 2023a), then enabling the curation or sculpting of high-quality datasets for training efficiency purposes (Paul et al., 2021) or improved performance (Liang et al., 2022). Typically, the goal is to identify mislabeled, hard, or ambiguous examples, with methods differing based on metrics including uncertainty (Swayamdipta et al., 2020; Seedat et al., 2022), logits (Pleiss et al., 2020), gradient norm (Paul et al., 2021), or variance of gradients (Agarwal et al., 2022). We note two key aspects: (1) we draw inspiration from their success in the fully supervised setting (where there are large amounts of labeled data) and bring the idea to the semi-supervised setting where we have unlabeled data but scarce labeled data; (2) many of the discussed supervised methods are only applicable to neural networks, relying on gradients or logits. Hence, they are not broadly applicable to any model class, such as boosting methods which are predominant in tabular settings (Borisov et al., 2021; Grinsztajn et al., 2022). This violates **P2: Model-agnostic data characterization**.

**Learning with Noisy Labels (LNL).** LNL typically operates in the supervised setting and assumes access to a large amount of labeled data. This contrasts the semi-supervised setting, where labeled data is scarce, and is used to output pseudo-labels for unlabeled data. Some LNL methods alter a loss function, e.g. adding a regularization term (Cheng et al., 2021; Wei et al., 2022b). Other methods select samples using a uni-dimensional metric, the most common being the small-loss criterion in the supervised setting (Xia et al., 2021). DIPS contrasts these approaches by taking into account both confidence and aleatoric uncertainty in its selection process. While the LNL methods have not been used in the semi-supervised setting previously, we repurpose them for this setting and experimentally highlight the value of the curation process of DIPS in Appendix C. Interestingly, pseudo-labeling can also be used as a tool in the supervised setting to relabel points identified as noisy by treating them as unlabeled (Li et al., 2019); however, this contrasts DIPS in two key ways: (1) data availability: these works operate *only* on large labeled datasets, whereas DIPS operates with a small labeled and large unlabeled dataset. (2) application: these works use pseudo-labeling as a tool for supervised learning, whereas DIPS extends the machinery of pseudo-labeling itself.

## 3 Background

We now give a brief overview of pseudo-labeling as a general paradigm of semi-supervised learning. We then highlight that the current formulation of pseudo-labeling overlooks the key notion of labeled data quality, which motivates our approach.

### 3.1 Semi-supervised learning via pseudo-labeling

Semi-supervised learning addresses the scarcity of labeled data by leveraging unlabeled data. The natural question it answers is: how can we combine labeled and unlabeled data to boost the performance of a model, compared to training on the small labeled data alone?

**Notation.** Consider a classification setting where we have a labeled dataset $\mathcal{D}_{\text{lab}} = \{(x_i, y_i)|i \in [n_{\text{lab}}]\}$ as well as an unlabeled dataset $\mathcal{D}_{\text{unlab}} = \{x'_j|j \in [n_{\text{unlab}}]\}$. We typically assume that $n_{\text{lab}} \ll n_{\text{unlab}}$. Moreover, the labels take values in $\{0, 1\}^C$, where $C$ is the number of classes. This encompasses both binary ($C = 2$) and multi-label classification. Our goal is to learn a predictive model $f : x \to y$ which leverages $\mathcal{D}_{\text{unlab}}$ in addition to $\mathcal{D}_{\text{lab}}$, such

that it performs better than a model trained on the small labeled dataset $\mathcal{D}_{\text{lab}}$ alone. For all $k \in [C]$, the $k$-th coordinate of $f(x)$ is denoted as $[f(x)]_k$. It is assumed to be in $[0, 1]$, which is typically the case after a softmax layer.

**Pseudo-labeling.** Pseudo-labeling (PL) is a powerful and general-purpose semi-supervised approach that answers the pressing question of how to incorporate $\mathcal{D}_{\text{unlab}}$ in the learning procedure. PL is an iterative procedure that spans $T$ iterations and constructs a succession of models $f^{(i)}$, for $i = 1, ..., T$. The result of this procedure is the last model $f^{(T)}$, which issues predictions at test time. The idea underpinning PL is to gradually incorporate $\mathcal{D}_{\text{unlab}}$ into $\mathcal{D}_{\text{lab}}$ to train the classifiers $f^{(i)}$. At each iteration $i$ of pseudo-labeling, two steps are conducted in turn. *Step 1:* The model $f^{(i)}$ is first trained with supervised learning. *Step 2:* $f^{(i)}$ then pseudo-labels unlabeled samples, a subset of which are selected to expand the training set of the next classifier $f^{(i+1)}$. The key to PL is the construction of these training sets. More precisely, let us denote $\mathcal{D}_{\text{train}}^{(i)}$ the training set used to train $f^{(i)}$ at iteration $i$. $\mathcal{D}_{\text{train}}^{(i)}$ is defined by an initial condition, $\mathcal{D}_{\text{train}}^{(1)} = \mathcal{D}_{\text{lab}}$, and by the following recursive equation: for all $i = 1, ..., T - 1$, $\mathcal{D}_{\text{train}}^{(i+1)} = \mathcal{D}_{\text{train}}^{(i)} \cup s(\mathcal{D}_{\text{unlab}}, f^{(i)})$, where $s$ is a selector function. Alternatively stated, $f^{(i)}$ outputs pseudo-labels for $\mathcal{D}_{\text{unlab}}$ at iteration $i$ and the selector function $s$ then selects a subset of these pseudo-labeled samples, which are added to $\mathcal{D}_{\text{train}}^{(i)}$ to form $\mathcal{D}_{\text{train}}^{(i+1)}$. Common heuristics define $s$ with metrics of confidence and/or uncertainty (e.g. greedy-PL (Lee et al., 2013), UPS (Rizve et al., 2021)). More details are given in Appendix A regarding the exact formulation of $s$ in those cases.

## 3.2 Overlooked aspects in the current formulation of pseudo-labeling

Having introduced the pseudo-labeling paradigm, we now show that its current formulation overlooks several key elements that will motivate our approach.

First, the selection mechanism $s$ only focuses on unlabeled data and ignores labeled data. This implies that the labeled data is considered "perfect". This assumption is not reasonable in many real-world settings where labeled data is noisy. In such situations, as shown in Fig. 1, noise propagates to the pseudo-labels, jeopardizing the accuracy of the pseudo-labeling steps (Nguyen et al., 2022a). To see why such propagation of error happens, recall that $\mathcal{D}_{\text{train}}^{(1)} = \mathcal{D}_{\text{lab}}$. Alternatively stated, $\mathcal{D}_{\text{lab}}$ provides the initial supervision signal for PL and its recursive construction of $\mathcal{D}_{\text{train}}^{(i)}$.

Second, PL methods do not update the pseudo-labels of unlabeled samples once they are incorporated in one of the $\mathcal{D}_{\text{train}}^{(i)}$. However, the intuition underpinning PL is that the classifiers $f^{(i)}$ progressively get better over the iterations, meaning that pseudo-labels computed at iteration $T$ are expected to be more accurate than pseudo-labels computed at iteration 1, since $f^{(T)}$ is the output of PL.

Taken together, these two observations shed light on an important yet overlooked aspect of current PL methods: the selection mechanism $s$ ignores labeled and previously pseudo-labeled samples. This naturally manifests in the asymmetry of the update rule $\mathcal{D}_{\text{train}}^{(i+1)} = \mathcal{D}_{\text{train}}^{(i)} \cup s(\mathcal{D}_{\text{unlab}}, f^{(i)})$, where the selection function $s$ is only applied to *unlabeled data* and ignores $\mathcal{D}_{\text{train}}^{(i)}$ at iteration $i + 1$.

## 4 DIPS: Data-centric insights for improved pseudo-labeling

In response to these overlooked aspects, we propose a new formulation of pseudo-labeling, DIPS, with the data-centric aim to characterize the usefulness of both *labeled* and *pseudo-labeled* samples. We then operationalize this framework with the lens of learning dynamics. Our goal is to improve the performance of *any* pseudo-labeling algorithm by selecting *useful* samples to be used for training.

### 4.1 A data-centric formulation of pseudo-labeling

Motivated by the asymmetry in the update rule of $\mathcal{D}_{\text{train}}^{(i)}$, as defined in Sec. 3.1, we propose DIPS, a novel framework which explicitly focuses on both labeled and pseudo-labeled samples. The key idea is to introduce a new selection mechanism, called $r$, while still retaining the benefits of $s$. For any dataset $\mathcal{D}$ and classifier $f$, $r(\mathcal{D}, f)$ defines a subset of $\mathcal{D}$ to be used for training in the current pseudo-labeling iteration. More formally, we define the new update rule (Eq. 1) for all $i = 1, ..., T - 1$ as:

$$\begin{cases} \mathcal{D}^{(i+1)} &= \mathcal{D}^{(i)} \cup s(\mathcal{D}_{\text{unlab}}, f^{(i)}) \quad \triangleright \text{Original PL formulation} \\ \mathcal{D}_{\text{train}}^{(i+1)} &= r(\mathcal{D}^{(i+1)}, f^{(i)}) \quad\quad\quad\quad \triangleright \text{DIPS selection} \end{cases} \tag{1}$$

Then, let $\mathcal{D}^{(1)} = \mathcal{D}_{\text{train}}^{(1)} = r(\mathcal{D}_{\text{lab}}, f^{(0)})$, where $f^{(0)}$ is a classifier trained on $\mathcal{D}_{\text{lab}}$ only. The selector $r$ selects samples from $\mathcal{D}^{(i+1)}$, producing $\mathcal{D}_{\text{train}}^{(i+1)}$, the training set of $f^{(i+1)}$.

This new formulation addresses the challenges mentioned in Sec. 3.2. Indeed, for any $j < i$, $r$ selects samples in $\mathcal{D}^{(j)}$ at any iteration $i$ (which includes labeled samples), since $\mathcal{D}^{(j)} \subset \mathcal{D}^{(i)}$ for all $i = 0, ..., T$. We investigate in Appendix C.1 alternative ways to incorporate the selector $r$, showing the value of the DIPS formulation.

Finally, notice that DIPS subsumes current pseudo-labeling methods via its selector $r$. To see that, we note current pseudo-labeling methods define an identity selector $r$, selecting all samples, such that for any $\mathcal{D}$ and function $f$, we have $r(\mathcal{D}, f) = \mathcal{D}$. Hence, DIPS goes beyond this status quo by permitting a non-identity selector $r$.

### 4.2 Operationalizing DIPS using learning dynamics

We now explicitly instantiate DIPS by constructing the selector $r$. Our key idea is to define $r$ using learning dynamics of samples. Before giving a precise definition, let us detail some context. Prior works in *learning theory* have shown that the learning dynamics of samples contain a useful signal about the nature of samples (and their usefulness) for a specific task (Arpit et al., 2017; Arora et al., 2019; Li et al., 2020). Some samples may be easier for a model to learn, whilst for other samples, a model might take longer to learn (i.e. more variability through training) or these samples might not be learned correctly during the training process. We build on this insight about learning dynamics, bringing the idea to the *semi-supervised setting*.

Our construction of $r$ has three steps. First, we analyze the learning dynamics of labeled and pseudo-labeled samples to define two metrics: (i) confidence and (ii) aleatoric uncertainty, which captures the inherent data uncertainty. Second, we use these metrics to characterize the samples as *Useful* or *Harmful*. Third, we select *Useful* samples for model training, which gives our definition of $r$.

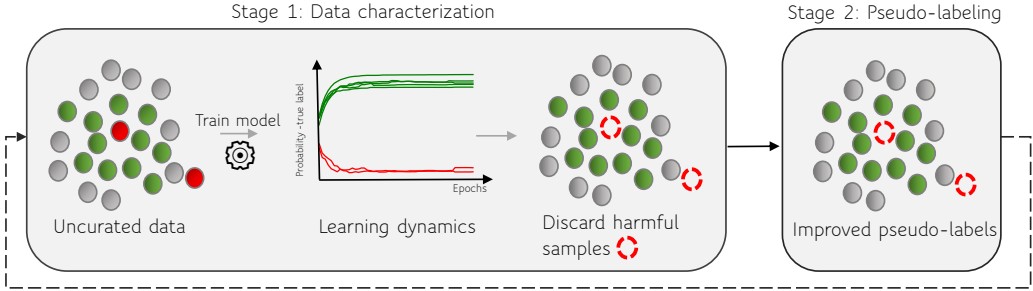

Figure 2: Stage 1 operationalizes DIPS by leveraging learning dynamics of individual labeled and pseudo-labeled samples to characterize them as *Useful* or *Harmful*. Only Useful samples are then kept for Stage 2, which consists of pseudo-labeling, using any off-the-shelf method.

For any $i$, we assume that the classifier $f^{(i)}$ at iteration $i$ of PL is trained in an iterative scheme (e.g. neural networks or XGBoost trained over iterations), which is ubiquitous in practice. This motivates and makes it possible to analyze the learning dynamics as a way to characterize individual samples. For clarity of presentation, we consider binary classification $(C = 2)$ and denote $f^{(i)} = f$.

At any pseudo-labeling iteration, $f$ is trained from scratch and goes through $e \in [E]$ different checkpoints leading to the set $\mathcal{F} = \{f_1, f_2, \ldots, f_E\}$, such that $f_e$ is the classifier at the $e$-th checkpoint. Our goal is to assess the learning dynamics of samples over these $E$ training checkpoints. For this, we define $H$, a random variable following a uniform distribution $\mathcal{U}_\mathcal{F}$ over the set of checkpoints $\mathcal{F}$. Specifically, given $H = h$ and a sample $(x, y)$, where $y$ is either a provided label ($x \in \mathcal{D}_{\text{lab}}$) or a pseudo-label ($x \in \mathcal{D}_{\text{unlab}}$), we define the correctness in the prediction of $H$ as a binary random variable $\hat{Y}_\mathcal{F}(x, y)$ with the following conditional: $P(\hat{Y}_\mathcal{F}(x, y) = 1 | H = h) = [h(x)]_y$ and $P(\hat{Y}_\mathcal{F}(x, y) = 0 | H = h) = 1 - P(\hat{Y}_\mathcal{F}(x, y) = 1 | H = h)$.

Equipped with a probabilistic interpretation of the predictions of a model, we now define our characterization metrics: (i) average confidence and (ii) aleatoric (data) uncertainty, inspired by (Kwon et al., 2020; Seedat et al., 2022).

**Definition 4.1** (Average confidence)**.** For any set of checkpoints $\mathcal{F} = \{f_1, ..., f_E\}$, the average confidence for a sample $(x, y)$ is defined as the following marginal:

$$\bar{\mathcal{P}}_\mathcal{F}(x, y) := P(\hat{Y}_\mathcal{F}(x, y) = 1) = \mathbb{E}_{H \sim \mathcal{U}_\mathcal{F}}[P(\hat{Y}_\mathcal{F}(x, y) = 1 | H)] = \frac{1}{E} \sum_{e=1}^{E} [f_e(x)]_y \qquad (2)$$

**Definition 4.2** (Aleatoric uncertainty)**.** For any set of checkpoints $\mathcal{F} = \{f_1, ..., f_E\}$, the aleatoric uncertainty for a sample $(x, y)$ is defined as:

$$v_{al, \mathcal{F}}(x, y) := \mathbb{E}_{H \sim \mathcal{U}_\mathcal{F}}[Var(\hat{Y}_\mathcal{F}(x, y) | H)] = \frac{1}{E} \sum_{e=1}^{E} [f_e(x)]_y (1 - [f_e(x)]_y) \qquad (3)$$

Intuitively, the aleatoric uncertainty for a sample $x$ is maximized when $[f_e(x)]_y = \frac{1}{2}$ for all checkpoints $f_e$, akin to random guessing. Recall aleatoric uncertainty captures the inherent data uncertainty, hence is a principled way to capture issues such as mislabeling.

This contrasts epistemic uncertainty, which is model-dependent and can be reduced simply by increasing model parameterization (Hüllermeier and Waegeman, 2021).

We emphasize that this definition of uncertainty is model-agnostic, satisfying **P2: Model-agnostic data characterization**, and only relies on having checkpoints through training. Hence, it comes for *free*, unlike ensembles (Lakshminarayanan et al., 2017). This fulfills **P3: Computationally cheap**. Moreover, it is applicable to any iteratively trained model (e.g. neural networks and XGBoost) unlike approaches such as MC-dropout or alternative training dynamic metrics using gradients (Paul et al., 2021) or logits (Pleiss et al., 2020). We note that confidence and uncertainty are defined as averages over *all* the training checkpoints, in order to capture the full learning trajectories of samples. We show in Appendix C.6 that ignoring earlier training checkpoints is suboptimal, highlighting that earlier checkpoints carry valuable signal for data characterization.

## 4.3 Defining the selector $r$: data characterization and selection

Having defined sample-wise confidence and aleatoric uncertainty, we characterize both labeled and pseudo-labeled samples into two categories, namely *Useful* and *Harmful*. Given a sample $(x, y)$, a set of training checkpoints $\mathcal{F}$, and two thresholds $\tau_{\text{conf}}$ and $\tau_{\text{al}}$, we define the category $c(x, y, \mathcal{F})$ as *Useful* if $\bar{\mathcal{P}}_{\mathcal{F}}(x, y) \geq \tau_{\text{conf}}$ and $v_{al,\mathcal{F}}(x, y) < \tau_{\text{al}}$, and *Harmful* otherwise.

Hence, a *Useful* sample is one where we are highly confident in predicting its associated label and for which we also have low inherent data uncertainty. In contrast, a harmful sample would have low confidence and/or high data uncertainty. Finally, given a function $f$ whose training led to the set of checkpoints $\mathcal{F}$, we can define $r$ explicitly by $r(\mathcal{D}, f) = \{(x, y) \mid (x, y) \in \mathcal{D}, c(x, y, \mathcal{F}) = Useful\}$.

## 4.4 Combining `DIPS` with *any* pseudo-labeling algorithm

We outline the integration of `DIPS` into *any* pseudo-labeling algorithm as per Algorithm 1 (see Appendix A). A fundamental strength of `DIPS` lies in its simplicity. The computational overhead is also small (no extra model training and storing checkpoints is not required) – i.e. satisfying **P3: Computationally cheap**, with only minimal overhead on forward passes. These are negligible compared to the pseudo-labeling process in general. Additionally, `DIPS` is easily integrated into *any* pseudo-labeling algorithm – i.e. **P1: Plug & Play**, making for easier adoption.

---
**Algorithm 1** Plug `DIPS` into *any* pseudo-labeler

---
1: Train a network, $f^{(0)}$, using the samples from $\mathcal{D}_{\text{lab}}$.
2: Plug-in `DIPS`: set $\mathcal{D}_{\text{train}}^{(1)} = \mathcal{D}^{(1)} = r(\mathcal{D}_{\text{lab}}, f^{(0)})$
3: **for** $t = 1..\text{T}$ **do**
4:     Initialize new network $f^{(t)}$
5:     Train $f^{(t)}$ using $\mathcal{D}_{\text{train}}^{(t)}$.
6:     Pseudo-label $\mathcal{D}_{\text{unlab}}$ using $f^{(t)}$
7:     Define $\mathcal{D}^{(t+1)}$ using the PL method's selector $s$
8:     Plug-in DIPS : Define $\mathcal{D}_{\text{train}}^{(t+1)} = r(\mathcal{D}^{(t+1)}, f^{(t)})$    ▷ Data characterization and selection, Sec. 4.3
9: **end for**
10: **return** $f_T$

---

## 5 Experiments

We now empirically investigate multiple aspects of DIPS[1]. We discuss the setup of each experiment at the start of each sub-section, with further experimental details in Appendix B.

1. **Characterization: Does it matter?** Sec. 5.1 analyzes the effect of not characterizing and selecting samples in all $\mathcal{D}_{\text{train}}^{(i)}$ in a synthetic setup, where noise propagates from $\mathcal{D}_{\text{lab}}$ to $\mathcal{D}_{\text{unlab}}$.

2. **Performance: Does it work?** Sec. 5.2 shows characterizing $\mathcal{D}_{\text{train}}^{(i)}$ using DIPS improves performance of various state-of-the-art pseudo-labeling baselines on 12 real-world datasets.

3. **Narrowing the gap: Can selection reduce performance disparities?** Sec. 5.2 shows that DIPS also renders the PL methods more comparable to one other.

4. **Data efficiency: Can similar performance be achieved with less labeled data?** Sec. 5.3 studies the efficiency of data usage of vanilla methods vs. DIPS on different proportions of labeled data.

5. **Selection across countries: Can selection improve performance when using data from a different country?** Sec. 5.4 assesses the role of selection of samples when $\mathcal{D}_{\text{lab}}$ and $\mathcal{D}_{\text{unlab}}$ come from different countries in a clinically relevant task.

6. **Other modalities**: Sec. 5.5 shows the potential to use DIPS in image experiments.

Hence, the experiments will demonstrate the core purpose of DIPS, which is to address the overlooked issue of labeled data quality in pseudo-labeling, validating DIPS as an effective framework to improve different pseudo-labelers.

### 5.1 Synthetic example: Data characterization and unlabeled data improve test accuracy

**Goal.** To motivate DIPS, we demonstrate (1) label noise harms pseudo-labeling and (2) characterizing and selecting data using DIPS consequently improves pseudo-labeling performance.

**Setup.** We consider a synthetic setup with two quadrants (Lee et al., 2023), as illustrated in Fig. 3b [2]. We sample data in each of the two quadrants from a uniform distribution, and each sample is equally likely to fall in each quadrant. To mimic a real-world scenario of label noise in $\mathcal{D}_{\text{lab}}$, we randomly flip labels with varying proportions $p_{\text{corrupt}} \in [0.1, 0.45]$

**Baselines.** We compare DIPS with two baselines **(i) Supervised** which trains a classifier using the initial $\mathcal{D}_{\text{lab}}$ **(ii) Greedy pseudo-labeling (PL)** (Lee et al., 2013) which uses both $\mathcal{D}_{\text{lab}}$ and $\mathcal{D}_{\text{unlab}}$. We use an XGBoost backbone for all the methods and we combine DIPS with PL for a fair comparison. We also instantiate our DIPS framework with other possible selectors from the noisy label literature namely the small-loss criterion (Xia et al., 2021), Fluctuation (Wei et al., 2022b) and FINE (Kim et al., 2021), thereby contrasting DIPS' selector based on learning dynamics with these other selectors.

**Results.** Test performance over 20 random seeds with different data splits and $n_{\text{lab}} = 100, n_{\text{unlab}} = 900$ is illustrated in Fig. 3c, for varying $p_{\text{corrupt}}$. It highlights two key elements. First, PL barely improves upon the supervised baseline. The noise in the labeled dataset

---

1. https://github.com/seedatnabeel/DIPS or https://github.com/vanderschaarlab/DIPS

2. Notice that the two quadrant setup satisfies the cluster assumption inherent to the success of semi-supervised learning (Chapelle et al., 2006).

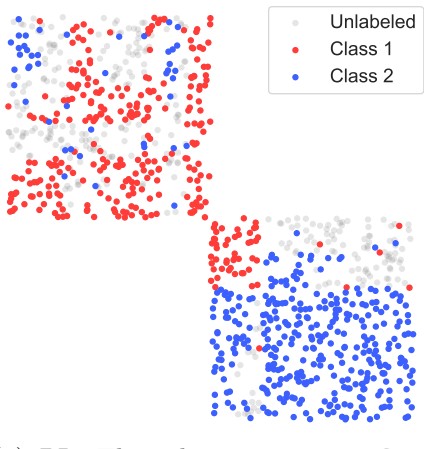

(a) PL: The inherent noise in $\mathcal{D}_{\text{lab}}$ propagates when assigning pseudo-labels.

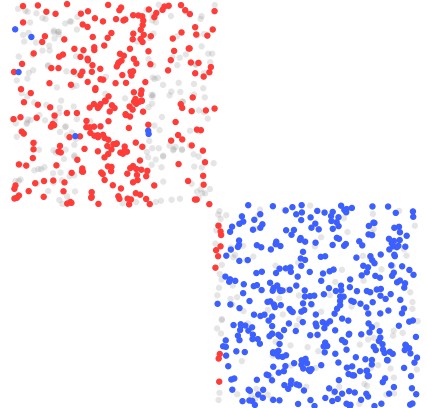

(b) PL+DIPS: DIPS mitigates the issue of noise by selecting useful samples.

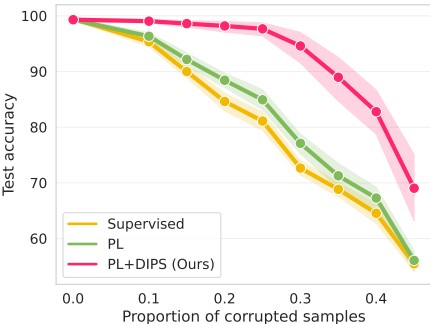

(c) DIPS selection mechanism significantly improves test performance under label noise.

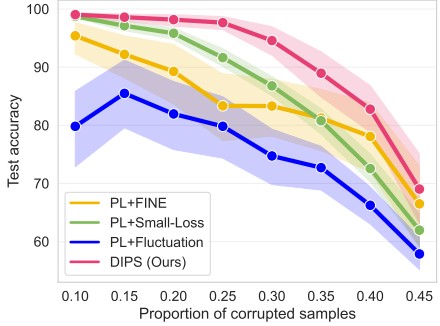

(d) DIPS selection mechanism outperforms other data-centric insights under label noise.

Figure 3: (a)-(b) The colored dots illustrate the selected labeled and pseudo-labeled samples for the last iteration of PL and PL+DIPS, with 30% label noise. Grey dots are unselected unlabeled samples. (c) Characterizing and selecting data for the semi-supervised algorithm yields the best results (epitomized by PL+DIPS) and makes the unlabeled data impactful. (d) Characterizing and selecting data via DIPS outperforms other data-centric mechanisms

propagates to the unlabeled dataset, via the pseudo-labels, as shown in Fig. 3a. This consequently negates the benefit of using the unlabeled data to learn a better classifier, which is the original motivation of semi-supervised learning. Second, DIPS mitigates this issue via its selection mechanism and improves performance by around **+20%** over the two baselines when the amount of label noise is around 30%. We also conduct an ablation study in Appendix C to understand when in the pseudo-labeling pipeline to apply DIPS.

**Takeaway.** The results emphasize the key motivation of DIPS: labeled data quality is central to the performance of the pseudo-labeling algorithms because labeled data drives the learning process necessary to perform pseudo-labeling. Hence, careful consideration of $\mathcal{D}_{\text{lab}}$ is crucial to performance.

## 5.2 DIPS improves different pseudo-labeling algorithms across 12 real-world tabular datasets.

**Goal.** We evaluate the effectiveness of DIPS on 12 different real-world tabular datasets with diverse characteristics (sample sizes, number of features, task difficulty). We aim to demonstrate that DIPS improves the performance of various pseudo-labeling algorithms. We focus on the tabular setting, as pseudo-labeling plays a crucial role in addressing data scarcity issues in healthcare and finance, discussed in Sec. 1, where data is predominantly tabular (Borisov et al., 2021; Shwartz-Ziv and Armon, 2022). Moreover, enhancing our capacity to improve models for tabular data holds immense significance, given its ubiquity in real-world applications. For perspective, nearly 79% of data scientists work with tabular data on a daily basis, compared to only 14% who work with modalities such as images (Kaggle, 2017). This underlines the critical need to advance pseudo-labeling techniques in the context of impactful real-world tabular data.

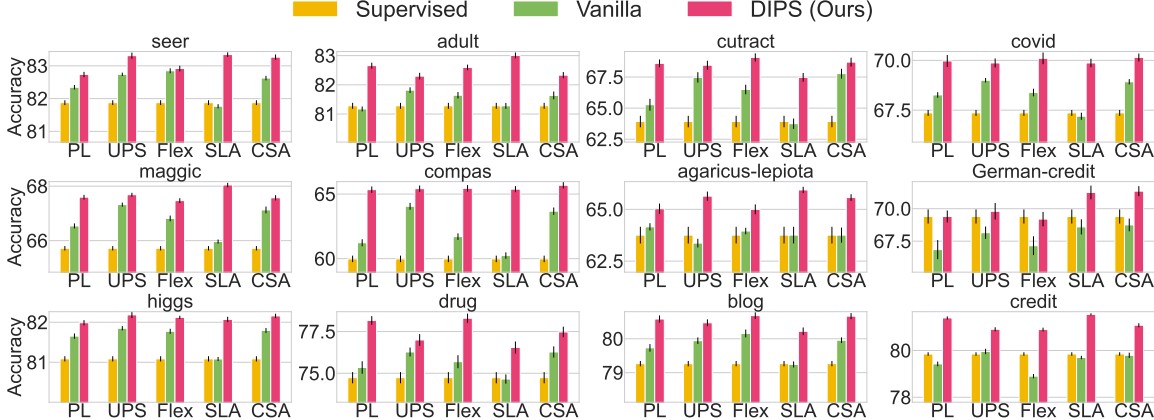

Figure 4: DIPS consistently improves the performance of all five pseudo-labeling methods across the 12 real-world datasets. DIPS also reduces the performance gap between the different pseudo-labelers.

**Datasets.** The tabular datasets are drawn from a variety of domains (e.g. healthcare, finance), mirroring Sec. 1, where the issue of limited annotated examples is highly prevalent. It is important to note that the vast majority of the datasets (10/12) are real-world datasets, demonstrating the applicability of DIPS and its findings in practical scenarios. For example, Covid-19 (Baqui et al., 2020), MAGGIC (Pocock et al., 2013), SEER (Duggan et al., 2016), and CUTRACT (Prostate Cancer PCUK, 2019) are medical datasets. COMPAS (Angwin et al., 2016) is a recidivism dataset. Credit is a financial default dataset from a Taiwan bank (Yeh and Lien, 2009). Higgs is a physics dataset (Baldi et al., 2014). The datasets vary significantly in both sample size (from 1k to 41k) and number of features (from 12 to 280). More details on the datasets can be found in Table 1, Appendix B.

**Baselines.** As baselines, we compare: (i) **Supervised** training on the small $\mathcal{D}_{\text{lab}}$, (ii) *five* state-of-the-art pseudo-labeling methods applicable to tabular data: **greedy-PL** (Lee et al., 2013), **UPS** (Rizve et al., 2021), **FlexMatch** (Zhang et al., 2021), **SLA** (Tai et al., 2021), **CSA** (Nguyen et al., 2022a). For each of the baselines, we apply DIPS as a plug-in to improve performance.

**Results** We report results in Fig. 4 across 50 random seeds with different data splits with a fixed proportion of $\mathcal{D}_{\text{lab}} : \mathcal{D}_{\text{unlab}}$ of 0.1:0.9. We note several findings from Fig. 4 pertinent to DIPS.

■ **DIPS improves the performance of almost all baselines across various real-world datasets.**
We showcase the value of data characterization and selection to improve SSL performance. We demonstrate that DIPS consistently boosts the performance when incorporated with existing pseudo-labelers. This illustrates the key motivation of our work: labeled data is of critical importance for pseudo-labeling and calls for curation, in real-world scenarios.

■ **DIPS reduces the performance gap between pseudo-labelers.**
Fig. 4 shows the reduction in variability of performance across pseudo-labelers by introducing data characterization. On average, we reduce the average variance across all datasets and algorithms from 0.46 in the vanilla case to 0.14 using DIPS. In particular, we show that the simplest method, namely greedy pseudo-labeling (Lee et al., 2013), which is often the worst in the vanilla setups, is drastically improved simply by incorporating DIPS, making it competitive with the more sophisticated alternative baselines. This result of equalizing performance is important as it directly influences the process of selecting a pseudo-labeling algorithm. We report additional results in Appendix C.2 where we replace the selector $r$ with sample selectors from the LNL literature, highlighting the advantage of using learning dynamics.

**Takeaway.** We have empirically demonstrated improved performance by DIPS across multiple pseudo-labeling algorithms and multiple real-world datasets.

### 5.3   DIPS improves data efficiency

**Goal.** In real-world scenarios, collecting labeled data is a significant bottleneck, hence it is traditionally the sole focus of semi-supervised benchmarks. The goal of this experiment is to demonstrate that data quality is an overlooked dimension that has a direct impact on data quantity requirements to achieve a given test performance for pseudo-labeling.

**Setup.** For clarity, we focus on greedy-PL and UPS as pseudo-labeling algorithms. To assess data efficiency, we

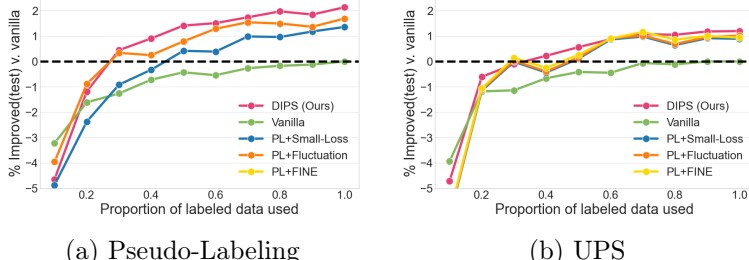

(a) Pseudo-Labeling           (b) UPS

Figure 5: DIPS (pink) improves data efficiency of vanilla methods (green), achieving the same level of performance with *60-70%* fewer labeled examples, as shown by the vertical dotted lines. The results (a) Pseudo-labeling and (b) UPS are averaged across datasets and show gains in accuracy vs. the maximum performance of the vanilla method. Additionally, DIPS selection generally provides additional efficiency gains over other possible selection mechanisms.

consider subsets of $\mathcal{D}_{\text{lab}}$ with size $p \cdot |\mathcal{D}_{\text{lab}}|$, with $p$ going from 0.1 to 1. We also instantiate

our `DIPS` framework with the small-loss criterion (Xia et al., 2021), Fluctuation (Wei et al., 2022b) and FINE (Kim et al., 2021) [3].

**Results.** The results in Fig. 5, averaged across datasets, show the performance gain in accuracy for all $p$ compared to the maximum performance of the vanilla method (i.e. when $p = 1$). We conclude that `DIPS` significantly improves the data efficiency of the vanilla pseudo-labeling baselines, between **60-70%** more efficient for UPS and greedy-PL respectively, to reach the same level of performance.

**Takeaway.** We have demonstrated that data quantity is not the sole determinant of success in pseudo-labeling. We reduce the amount of data needed to achieve a desired test accuracy by leveraging the selection mechanism of `DIPS`. This highlights the significance of a multi-dimensional approach to pseudo-labeling, where a focus on quality reduces the data quantity requirements.

### 5.4 `DIPS` improves performance of cross-country pseudo-labeling

**Goal.** To further assess the real-world benefit of `DIPS`, we consider the clinically relevant task of improving classifier performance using data from hospitals in different countries.

**Setup.** We assess a setup using Prostate cancer data from the UK (CUTRACT (Prostate Cancer PCUK, 2019)) to define $(\mathcal{D}_{\text{lab}}, \mathcal{D}_{test})$, which is augmented by $\mathcal{D}_{\text{unlab}}$, from US data (SEER (Duggan et al., 2016)). While coming from different countries, the datasets have interoperable features and the task is to predict prostate cancer mortality.

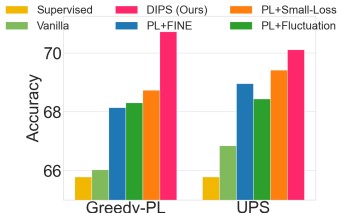

Figure 6: Curation of $\mathcal{D}_{\text{lab}}$ permits us to better leverage a cross-country $\mathcal{D}_{\text{unlab}}$

We leverage the unlabeled data from the US to augment the small labeled dataset from the UK, to improve the classifier when used in the UK (on $\mathcal{D}_{test}$). We also instantiate our `DIPS` framework with the small-loss criterion (Xia et al., 2021), Fluctuation (Wei et al., 2022b) and FINE (Kim et al., 2021)

**Results.** Fig. 6 illustrates that greedy-PL and UPS benefit from `DIPS`'s selection of labeled and pseudo-labeled samples, resulting in improved test performance. Hence, this result underscores that ignoring the labeled data whilst also naively selecting pseudo-labeled samples simply using confidence scores (as in greedy-PL) yields limited benefit. We provide further insights into the selection and gains by `DIPS` in Appendix C.

**Takeaway.** `DIPS`'s selection mechanism improves performance when using semi-supervised approaches across countries.

### 5.5 `DIPS` works with other data modalities

**Goal.** While `DIPS` is mainly geared towards the important problem of pseudo-labeling for tabular data, we explore an extension of `DIPS` to images, highlighting its versatility.

**Setup.** We investigate the use of `DIPS` to improve pseudo-labeling for CIFAR-10N (Wei et al., 2022a). With realism in mind, we specifically use this dataset as it reflects noise in image data stemming from real-world human annotations on M-Turk, rather than synthetic noise models (Wei et al., 2022a).

---

3. PL+FINE is below -6% and hence not shown for clarity of visuals

We evaluate the semi-supervised algorithm FixMatch (Sohn et al., 2020) with a WideResNet-28 (Zagoruyko and Komodakis, 2016) for $n_{lab} = 1000$ over three seeds. FixMatch combines pseudo-labeling with consistency regularization, hence does not apply to the previous tabular data-focused experiments.

We incorporate DIPS as a plug-in to the pseudo-labeling component of FixMatch.

**Results.** Fig. 7 showcases the improved test accuracy for FixMatch+DIPS of **85.2%** over vanilla FixMatch of **82.6%**. A key reason is that DIPS discards harmful mislabeled samples from $\mathcal{D}_{\text{lab}}$, with example images shown in Fig. 8b. Fig. 7 also compares DIPS with the small-loss criterion, showing the superior performance of DIPS' selection mechanism. Furthermore, Table 8a shows the addition of DIPS improves time efficiency significantly, reducing the final computation time by **8 hours**.

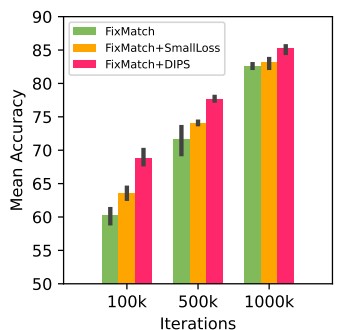

Figure 7: DIPS improves Fix-Match on CIFAR-10N, and also outperforms the selector based on the small-loss heuristic.

We show additional results for CIFAR-100N in Appendix C.4 and for other image datasets in Appendix C.4.4.

**Takeaway.** DIPS is a versatile framework that can be extended to various data modalities.

| Test acc (%) | FM + DIPS | FixMatch (FM) |
|:---:|:---:|:---:|
| 65 | **2.3** $\pm$ 0.4 | 8.0 $\pm$ 2.0 |
| 70 | **4.6** $\pm$ 0.5 | 16.4 $\pm$ 3.26 |
| 75 | **10.8** $\pm$ 0.7 | 26.5 $\pm$ 1.9 |
| 80 | **27.8** $\pm$ 0.8 | 35.8 $\pm$ 0.9 |
| 85 | **38.5** $\pm$ 0.3 | N.A. |

(a)

Noisy label: automobile  True label: airplane
Noisy label: deer  True label: horse
Noisy label: dog  True label: deer
Noisy label: truck  True label: automobile

(b)

Figure 8: (a) DIPS improves the time efficiency (hours reported on a v100 GPU) of FixMatch, by 1.5-4X for the same performance (↓ better). (b) Examples of mislabeled samples in CIFAR-10N discarded by DIPS. We note the incorrect labels and ideal ground-truth labels.

## 6 Discussion

We propose DIPS, a plugin designed to improve *any* pseudo-labeling algorithm. DIPS builds on the key observation that the quality of labeled data is overlooked in pseudo-labeling approaches, while it is the core signal that renders pseudo-labeling possible. Motivated by real-world datasets and their inherent noisiness, we introduce a cleaning mechanism that operates both on labeled and pseudo-labeled data. We showed the value of taking into account labeled data quality – by characterizing and selecting data we improve test performance for various pseudo-labelers across 18 real-world datasets spanning tabular data and images.

## Acknowledgments and Disclosure of Funding

NS is supported by the Cystic Fibrosis Trust and NH by Illumina. This work was supported by Azure sponsorship credits granted by Microsoft's AI for Good Research Lab.

## Broader Impact Statement

In this work, we delve into the essential yet often neglected aspect of labeled data quality in the application of pseudo-labeling, a semi-supervised learning technique. Our key insights stem from a data-centric approach that underscores the role of labeled data quality - a facet typically overlooked due to the default assumption of labeled data being "perfect". In stark contrast to the traditional, algorithm-centric pseudo-labeling literature which largely focuses on refining pseudo-labeling methods, we accentuate the critical influence of the quality of labeled data on the effectiveness of pseudo-labeling.

By way of introducing the `DIPS` framework, our work emphasizes the value of characterization and selection of labeled data, consequently improving any pseudo-labeling method. Moreover, akin to traditional machine learning problems, focusing on labeled data quality in the context of pseudo-labeling promises to lessen risks, costs, and potentially detrimental consequences of algorithm deployment. This perspective opens up many avenues for applications in areas where labeled data is scarce or expensive to acquire, including but not limited to healthcare, social sciences, autonomous vehicles, wildlife conservation, and climate modeling scenarios. Our work underscores the need for a data-centric paradigm shift in the pseudo-labeling landscape.

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
