# OpenReview forum: "You can't handle the (dirty) truth: Data-centric insights improve pseudo-labeling"
_DMLR — Accepted by DMLR_

### Review · Reviewer_JdaE · 2024-04-18

**Recommendation:** 4
**Confidence:** 2

**Summary Of Contributions:**

Semi-supervised learning involves a learning from a small labeled dataset and a larger unlabeled dataset. One approach is pseudolabeling, where at each iteration, 1) a model is trained, 2) the model is used to pseudolabel the unlabeled data, and 3) some of the pseudolabeled data is selected to be incorporated into the next iteration's training dataset. However, existing methods ignore the fact that labeled data can often be ambiguous or mislabeled. When not accounted for, pseudolabeling can thus compound the effects of low-quality labeled data. This paper proposes a simple modification on top of any pseudolabeling algorithm called DIPS, which involves filtering each iteration's training dataset based on confidence and aleatoric uncertainty as measured by training dynamics. DIPS is able to improve performance on 18 datasets over a variety of standard pseudolabeling approaches.

**Strengths:**

Significance:
- DIPS applies to and improves on a variety of iterative pseudolabeling methods, making its contribution fairly significant to the area of semi-supervised learning. Also, my understanding from the experiments was that there was no significant tuning of thresholds per dataset, so the method seems to be simple and broadly useful.

Relation to prior work:
- Paper discusses how DIPS can improve on other pseudolabeling methods, and also compares to other settings for data selection, such as active learning.

Relevance:
- DIPS calls attention to the fact that even labeled datasets are not perfect. This is quite important more broadly in ML research, as many recent methods incorporate some aspect of self-improvement/self-training that relies on a sufficient-quality reference dataset due to data bottlenecks.

Quality:
- DIPS is simple and does not incur any additional costs. It is able to improve performance across many datasets, and ablations justify the choice of selection metric well. Additional experiments on several modalities as well as the country distribution shift experiment help confirm that DIPS is performing as expected.

Clarity:
- Paper is well-written with key takeaways highlighted, as well as the technical modification of DIPs clearly explained.

Ethical/Social Implications:
- No concerns.

**Audience:**

Yes

**Broader Impact Concerns:**

No concerns.

**Claims And Evidence:**

Yes.

**Datasets And Benchmarks:**

N/A

**Extended Submissions:**

N/A

**Limitations:**

**General comments:**

1. I don't find the individual modification to the pseudolabeling procedure or the proposed confidence/uncertainty metric to be that novel. However, when put together it appears to offer significant improvements, so I am not very concerned about this in my overall recommendation.
2. One advantage of DIPS is that it does not incur any additional costs. However, if DIPS were to be applied in a standard supervised data selection setting that was not iterative, it would suffer from the same drawbacks, e.g., "having to train a model to select data to train a model". Therefore, I feel that the computational advantage is purely based on the setting rather than being inherently interesting.

**Questions about relationship between DIPS and corruption rate of labeled data:**

3. Let's suppose that we have prior knowledge of how mislabeled the true data is, such as a 10% error rate. Ideally, should we be able to translate this into an optimal $\tau$? For instance, if the labeled data is 100% clean, should the $\tau$ always be set to be very generous such that very few samples are filtered out?
4. Alternatively, does DIPS have advantages in the clean labeled dataset setting? By selecting high confidence, high certainty points, one can imagine that DIPS is performing some sort of easy-to-hard curriculum learning even in the noise-free setting.
5. Are there any available statistics on the quality of labels for the datasets considered in this paper? I noticed for German-credit, DIPS only slightly improves over the supervised setting sometimes and that standard pseudolabeling does worse than supervised. Do you have a potential explanation or more context you can provide on this dataset?


**Technical questions:**

6. It is not clear how the adaptive $\tau_{al}$ is motivated. It seems from Figure 21 that the adaptive threshold outperforms both static thresholds; is it overcoming some fundamental challenge of using static thresholds then?
7. With regards to the dynamics plot in Figure 9, are there any cases of samples that start out with low confidence at the initial checkpoint but end up significantly improving? (in contrast to the plots in Fig 9 where the useful and harmful samples appear to be clearly separated). I imagine that these points may not be considered useful since the average confidence over the checkpoints is not that high. It also feels like these metrics should not regard checkpoints as IID but rather some timeseries, but I'm unsure if that makes a big difference in performance.

**Requested Changes:**

* [Would strengthen] Discuss how thresholds for DIPS should be chosen, justifying adaptive versus static thresholds
* [Would strengthen] Add more discussion/synthetic experiments on if DIPS has any additional advantages in a clean label setting.
* [Would strengthen] If any statistics on data quality of the evaluated datasets are known, it would be helpful to provide them.
* Answering my other questions not mentioned here (Qs 3-7)

---

### Review · Reviewer_7MNS · 2024-04-25

**Recommendation:** 3
**Confidence:** 2

**Summary Of Contributions:**

This paper proposes Data-centric Insights for Pseudo-labeling with Selection (DIPS) and validate the effectiveness of DIPS through sufficient experiments. The DIPS adds an additional step in the recursive step of pseudo-labeling by reconsidering all the selected data (labeled data and pseudo-labeled data) with two selection criteria based on existing trained model trajectory. Conceptually, DIPS combines two settings: 1. noisy label and 2. semi-supervised learning. Experiments shows that DIPS can achieve overall best performance on 12 small tabular datasets and improvements in different domains and modalities.

**Strengths:**

- This paper is well structured and clear written.
- The performance is good in most settings. The comprehensive experiments makes most arguments are convincing.

**Audience:**

Yes

**Broader Impact Concerns:**

The author may want to mention that the advantage of their methods comes with a price of computation. For example, repeated evaluation over the entire unlabeled datasets if there are no further treatment.

**Claims And Evidence:**

Yes.

**Datasets And Benchmarks:**

Not apply.

**Extended Submissions:**

This is not an extended submission.

**Limitations:**

The key issue DIPS tries to tackle is the noise in both annotated label and pseudo-label.  This issue is somehow discussed in the Appendix C.2 with different selector (or other methods handling the label noise).

The main results, however, are majorly based on (semi-)supervised methods without special treatment against the label noise. From my perspective, those experiments can only show that the general hybrid framework of pseudo-labeling and label noise reduction works well, but not able to fully justify the effectiveness of the two data-centric "insights" derived from the learning dynamics.

**Requested Changes:**

If the comparison of selectors (small-loss, Fluctuation, FINE) can be added to major results tables/figures, such as Figure 3 (c), Figure 5, Figure 6 and Figure 7. Therefore, the effectiveness of two data-centric insights can be further elaborated under small tabular datasets, cross-country setting, and different modality.

---

### Review · Reviewer_sFzt · 2024-04-26

**Recommendation:** 4
**Confidence:** 3

**Summary Of Contributions:**

The paper challenges the assumption in Pseudo-labeling that labeled samples are always correct and perfect, proposing a new framework to enhance labeling performance. It introduces a hybrid condition for selecting samples to be labeled, based on average confidence and predicted uncertainty across training checkpoints. This approach demonstrates improved labeling accuracy and data usage efficiency. The framework is designed to be adaptable to any PL framework and model architecture, with empirical evidence showcasing its effectiveness on various real-world datasets.

New Knowledge:
* The new idea proposed is the mixed condition involving confidence and uncertainty over training process.

**Strengths:**

* The paper is well-structured and written in a clear and easy-to-understand manner. It verifies motivations with noise propagation and emphasizes that this problem is crucial to be studied.
* The proposed framework, DIPS, is computationally efficient as it leverages a single training process to extract learning dynamics. It is adaptable to other PL algorithms and models.
* The authors effectively demonstrate the advantages of the framework in various settings, especially cross-modalities and cross-distribution settings, as well as in real-world datasets.

**Audience:**

Yes

**Broader Impact Concerns:**

I didn't observe any ethical concerns related to the work.

**Claims And Evidence:**

Their claims are clear and convincing.

**Datasets And Benchmarks:**

The data collection and related information are sufficiently detailed.

**Extended Submissions:**

I believe it meets the eligibility criteria.

**Limitations:**

* It is unclear how to calculate the "average" in learning dynamics over checkpoints. Should information from earlier or later epochs be considered?
* The threshold selection also needs better clarification.
* The selector considers useful samples at the end. What about harmful samples? Is there a datapoint that is always considered harmful?
* It's important to explore the thresholds $\tau_{conf}$ and $\tau_{al}$ and how they impact the labeling performance and coverage.

**Requested Changes:**

* Using "average" in learning dynamics requires further justification.
* What are the changes across the thresholds $\tau_{conf}$ and $\tau_{al}$?